



# Measurement report: Investigation of Optical Properties of Different Fuels Diesel Exhaust by an Atmospheric Simulation Chamber experiment

Silvia G. Danelli[1], Lorenzo Caponi[1], Marco Brunoldi[2,3], Matilde De Camillis[1], Dario Massabò[2,3], Federico Mazzei[2,3], Tommaso Isolabella[2,3], Annalisa Pascarella[4,5], Paolo Prati[2,3], Matteo Santostefano[1], Francesca Tarchino[6], Virginia Vernocchi[2], Paolo Brotto[1]

[1]PM_TEN Srl, Genoa, 16123, Italy
[2]INFN, Genoa Section, Genoa, 16146, Italy
[3]Department of Physics, University of Genoa, Genoa, 16146, Italy
[4]BEES Srl, Genoa, 16121, Italy
[5]Istitute for the Application of Calculus (IAC) – CNR, Rome, 00185, Italy
[6]SIGE Srl, Genoa, 16161, Italy

*Correspondence to*: Federico Mazzei (federico.mazzei@ge.infn.it)

**Abstract.** This study investigates the optical properties and variability of the mass absorption coefficient (MAC) of carbonaceous aerosols produced by the combustion of different fuels. Emissions were also characterized in terms of particle size distribution and concentrations of elemental (EC) and organic carbon (OC). Experiments were conducted in an atmospheric simulation chamber with a soot generator fueled with propane and a commercial diesel engine running on regular diesel and Hydrotreated Vegetable Oil (HVO). Different methods of sampling and analyzing carbonaceous aerosols were evaluated, focusing on workplace environments. The EC:TC (total carbon) ratios were found to be around 0.7 for propane, 0.15 for diesel, and 0.4 for HVO, indicating a higher proportion of OC in the diesel and HVO samples. Fresh soot particles showed monomodal log-normal distributions with peaks varying based on the fuel type and combustion process, with propane particles exhibiting a peak at larger particle sizes compared to HVO and diesel. The optical properties revealed that the MAC values varied across different fuel exhausts. Diesel combustion produced more light-absorbing particles compared to propane and HVO, with MAC values measured between 870 and 635 nm ranging from $6.2 \pm 0.5$ to $9.4 \pm 0.4$ m² g⁻¹ for commercial diesel, $5.2 \pm 0.5$ to $7.8 \pm 1.1$ m² g⁻¹ for propane, and $5.8 \pm 0.2$ to $8.4 \pm 0.6$ m² g⁻¹ for HVO.

## 1 Introduction

Understanding of the processes involving carbonaceous aerosols, which constitute 20% to 50% of total aerosol mass in the atmosphere (Kanakidou et al., 2005, Putaud et al., 2010) is crucial for both climate and human health. Actually, carbonaceous aerosol produced during the incomplete combustion of biomass and fossil fuels significantly impact clime (Ackerman et al., 2000; Menon et al., 2002; Quinn et al., 2008; Ramanathan and Carmichael, 2008; Bond et al., 2013) and human health (Kanaya



et al., 2008, Sandrini at al., 2014, Pope et al., 2002; Anenberg et al., 2010; Gan et al., 2011; Cassee et al., 2013; Lelieveld et al., 2015).

The combustion-related absorbing particles are commonly referred to as black carbon (BC) when analyzed optically (Petzold
et al., 2013) and as elemental carbon (EC) when characterized thermally (Bond and Bergstrom, 2006). Nevertheless, BC and EC often yield different concentration values (Massabò and Prati, 2021). Another important fraction of the by-product of combustion processes is organic carbon (OC). OC refers to the non-refractory fraction of carbonaceous aerosols, which can include numerous organic species. Among these, the light-absorbing species are known as Brown Carbon (BrC) (Moosmüller et al., 2009). The comparability of different thermal-optical protocols for OC and EC measurements (Cavalli et al., 2010,
Giannoni et al., 2016) and the comparability of BC and EC measurements (Reisinger et al., 2008, Salako et al., 2012) remain active areas of research.

Soot particles, carbonaceous particles that are a by-product of the incomplete combustion of fossil fuels and/or biomass burning (Nordmann et al., 2013; Moore et al., 2014), are a significant component of anthropogenic particulate matter (PM), especially in urban areas, and are emitted by traffic, domestic stoves, industrial chimneys, and diesel engines (Weijers et al., 2011). Diesel
engine exhaust, a complex mixture of gases, vapors, and fine particles, were classified in 2012 by the IARC as carcinogenic to humans (IARC category 1) and EC, a significant component of these emissions, is indicated as a common marker of exposure. For instance, Directive (EU) 2019/130, which amends Directive 2004/37/EC on the protection of workers from risks related to exposure to carcinogens or mutagens at work, sets binding occupational exposure limit values for diesel engine exhaust emissions at 0.05 mg m$^{-3}$, measured as EC.

In September 2021, the WHO updated its Air Quality Guidelines based on a systematic review of scientific evidence on the health effects of air pollution and, while data had been considered not sufficient to recommend specific AQG levels for BC and EC, the guidelines emphasized the need for further research and mitigation strategies to face health concerns. Lastly, the new Directive (EU) 2024/2881 on ambient air quality and cleaner air for Europe, incorporates the latest scientific evidence, including the updated WHO guidelines and, among various pollutants agents, it mandates the measurement of BC and EC in
both rural and urban locations to support scientific understanding of their health and environmental impacts.

Understanding the properties and behavior of soot particles in the atmosphere, such as their spectral optical properties, is essential to fully assess their adverse effects and to properly define some of the methodologies used for their determination. The quantitative definition of the light absorbing properties of atmospheric aerosols is usually expressed by the mass absorption coefficient, MAC, which was first introduced by Putaud et al., 2010. MAC is the light absorption cross section normalized to
the mass of a given species (e.g., EC/BC and/or BrC) of aerosol particles and it is given in units of (m$^2$ g$^{-1}$) and defined as

$$MAC(\lambda) = \frac{b_{abs(\lambda)}}{m},$$   (1)

where $b_{abs}(\lambda)$, units of m$^{-1}$, is defined as the absorption coefficient at a specific wavelength and $m$ is the mass concentration of the specific absorbing aerosol fraction.





Commonly accepted MAC values for freshly emitted BC are around $7.5 \pm 1.2$ m² g⁻¹ at λ = 550 nm (Bond & Bergstrom 2006,

Bond et al. 2013), with a recent study suggesting $8.0 \pm 0.7$ m² g⁻¹ (Liu et al. 2019). However, atmospheric measurements show a wide range of MAC values for BC-containing aerosols, from 5.5 to 45.9 m² g⁻¹ at λ = 550 nm and 4.2 to 19.9 m² g⁻¹ at λ = 637 nm (Genberg et al., 2013, Zanatta et al., 2016).

In this frame, we investigated the variability of MAC of carbonaceous aerosols produced under different fuel combustion conditions. Additionally, we evaluated and compared different methods of sampling and analyzing carbonaceous aerosols,

with a particular focus on those used in workplace environments. Soot particles were collected using both an environmental monitoring sampler and personal air samplers, which are typically used to monitor workers' exposure to dust in various occupational settings. This selection reflects the increasing awareness and regulatory focus on the health impacts of diesel exhaust exposure in the workplace. An effective monitoring helps ensure compliance with environmental regulations and occupational safety standards and these samplers were included to cover all conditions and tools commonly employed for

sampling carbonaceous aerosols in regulated environments. Indeed, this work is part of the CALIPSO project (Airborne Carbon: Limits, Impact, Protocols, and Operational Standards), funded by the Italy Liguria Region's PR FESR 2021–2027 program, which aims to evaluate and compare different methods of sampling and analysing carbonaceous aerosols, especially in workplaces.

The experiments were conducted inside an atmospheric simulation chamber (ASC) alternatively connected to a soot generator

and a commercial diesel engine running on regular diesel and Hydrotreated Vegetable Oil (HVO). The use of an ASC allows for controlled, realistic environmental conditions, offering a compromise between laboratory and field experiments by providing quasi-realistic conditions without the variability of field measurements (Finlayson-Pitts and Pitts, 2000; Becker, 2006). Some examples of recent ASC applications studying the physic-chemical and optical properties of different aerosol types are Caponi et al., 2017, Kumar et al., 2018, Hu et al., 2021 and Vernocchi et al. 2022.

## 2 Materials and methods

Experiments took place at the ChAMBRe (Chamber for Aerosol Modelling and Bio-aerosol Research) facility, located at the Physics Department of the University of Genoa and managed jointly with the Genoa Division of the National Institute of Nuclear Physics (INFN). ChAMBRe is a stainless-steel chamber, with a volume of about 2.2 m³. A detailed description can be found in Massabò et al., 2018; Danelli et al., 2021, Vernocchi et al., 2023. Temperature, pressure, and humidity, as well as

the gaseous and aerosol content, can be continuously monitored. The homogeneity of the mixture is ensured by a fan placed at the bottom of the chamber which allows a mixing time of about 180 s, with a fan rotating speed of 1.6 revolutions per second (Massabò et al. 2018). Between consecutive experiments, ChAMBRe can be evacuated to $10^{-5}$ mbar using a composite pumping system, which includes a TRIVAC® D65B rotary pump, a RUVAC WAU 251 root pump, and a Turbovac 1000, all from Leybold Vacuum. Before and during the experiments, ambient air enters the chamber through a five-stage filtering and

purifying inlet, which includes a HEPA filter (model PFIHE842; NW25/40 inlet/outlet – 25/55 SCFM; 99.97% efficient at 0.3



μm). The chamber is equipped with several flanges to allow a large panel of instruments to be connected to measure online and offline gaseous composition and aerosol concentration and properties inside the volume: an overview of the techniques used to characterize soot particles is reported in the following Sections.

A total of 10 experiments were performed to investigate the properties of carbonaceous aerosols and observe the changes in

MAC by varying different fuel types and combustion conditions, as summarized in Table 1. Soot particles were introduced into ChAMBRe, as detailed in Sect. 2.1, and were monitored using online instrumentation and sampled for offline analysis at various time intervals to investigate the concentration of emitted particles, their size distribution, and optical properties. Both optical and thermal-optical techniques were used for measurements. The combination of optical and thermal-optical analyses offers several advantages, such as the ability to determine the MAC value (Janssen et al., 2011; Gentner et al., 2012; Robinson

et al., 2007) of the particulate matter in specific conditions and to improve the accuracy of OC/EC separation (Pio et al., 2011).



**Table 1 Full list of the experiments analyzed in the present work. Additional information includes the start and total duration of the sampling and measurement interval for soot particles.**

| Date | Experiment | Type of Fuel – Soot particles source | Start of the experiment | Meas. Interval [min] |
|---|---|---|---|---|
| 17/07/2024 | P1a | Propane – MISG | 3 min after the injection | 110 |
| 17/07/2024 | P1b | Propane – MISG | 2 hours after the injection | 190 |
| 18/07/2024 | P2 | Propane – MISG | 3 min after the injection | 120 |
| 19/07/2024 | D3a | DIESEL - 65230 – 6 kW -Hyundai | 3 min after the injection | 120 |
| 19/07/2024 | D3b | DIESEL - 65230 – 6 kW -Hyundai | 2 hours and 30 minutes after the injection | 110 |
| 22/07/2024 | D4a | DIESEL - 65230 – 6 kW -Hyundai | 3 min after the injection | 190 |
| 22/07/2024 | D4b | DIESEL - 65230 – 6 kW -Hyundai | 4 hours after the injection | 150 |
| 23/07/2024 | H5a | HVO - 65230 – 6 kW -Hyundai | 3 min after the injection | 80 |
| 23/07/2024 | H5b | HVO - 65230 – 6 kW -Hyundai | 5 hours after the injection | 130 |
| 24/07/2024 | H6 | HVO - 65230 – 6 kW -Hyundai | 3 min after the injection | 100 |
| 25/07/2024 | H7a | HVO - 65230 – 6 kW -Hyundai | 30 min after the injection | 70 |
| 25/07/2024 | H7b | HVO - 65230 – 6 kW -Hyundai | 4 hours after the injection | 130 |
| 26/07/2024 | H8 | HVO - 65230 – 6 kW -Hyundai | 3 min after the injection | 170 |
| 16/12/2024 | P9a | Propane – MISG | 3 min after the injection | 60 |
| 16/12/2024 | P9b | Propane – MISG | 1 hour and 30 min after the injection | 120 |
| 17/12/2024 | P10a | Propane – MISG | 3 min after the injection | 60 |
| 17/12/2024 | P10b | Propane – MISG | 1 hour and 20 min. after the injection | 180 |

## 2.1 Particle Generation

Injections of fresh soot particles inside ChAMBRe were performed alternatively by a mini-inverted soot generator (MISG; Argonaut Scientific Corp., Edmonton, AB, Canada; model MISG-2), fueled with propane and by a 12 HP 4-stroke diesel engine (Electrical Generator 65230 – 6 kW - Hyundai), fueled alternatively with regular fossil diesel and HVO. HVO is a renewable biofuel made by hydrotreating vegetable oils, animal fats or waste oils. It is considered environmentally friendly because it is free of aromatics, oxygen, and sulfur, and can potentially reduce emissions compared to conventional diesel

(Zeman et al., 2019, Orliński, P. et al., 2024). HVO meets diesel fuel standards, allowing it to be used in existing engines and infrastructure without modifications.

The MISG is an inverted-flame burner often considered as an ideal soot source (Stipe et al., 2005, Moallemi et al., 2019 and references therein). The MISG can be operated with different fuels, such as ethylene (Kazemimanesh et al., 2019) and propane




(Moallemi et al., 2019; Bischof et al., 2019). A comprehensive characterization of the MISG soot particles to perform
experiments in atmospheric simulation chambers is reported in Vernocchi et al., 2022.

The efficiency of the combustion process (i.e. fuel lean/rich) can be expressed in terms of the global equivalence ratio ($\varphi$), which is the ratio of the actual fuel-to-air ratio to the stoichiometric fuel-to-air ratio, as follows:

$$\varphi = \frac{(m_F/m_A)}{(m_F/m_A)_{st}}, \qquad\qquad (2)$$

where $(m_F/m_A)$ and $(m_F/m_A)_{st}$ respectively are the actual and the stoichiometric fuel-to-air ratios. The fuel-to-air ratio is
the inverse of the air-to-fuel ratio (AFR), which is the ratio of air to fuel mass. The stoichiometric AFR value for propane is 15.64 (inverse value is 0.064). In this work, only fuel-lean conditions were used (i.e. $\varphi < 1$) considering that low fuel-to-air ratios are expected to produce particles with a high fraction of EC (Mamakos et al., 2013).

In this study, the MISG was fueled with propane with fixed air-to-fuel flow ratio, based on Vernocchi et al., 2022. Operative conditions selected for propane combustion are reported in Table 2.

Diesel exhaust emissions were produced by the engine of the electrical generator and introduced into ChAMBRe using the same experimental layout adopted for the MISG and fully described in Vernocchi et al., 2022. The Hyundai Electrical Generator 65230 was connected to ChAMBRe using a connection line made with Swagelok Adaptors (size 3/4''; 19.05) and ISO-K flanges (16 mm diameter) to prevent any possible leaks. The diesel generator is compliant with the Stage V EU normative, introduced in 2019 to reduce harmful pollutants like nitrogen oxides ($NO_x$) and particulate matter (PM) from diesel-
powered equipment (Regulation EU 2016/1628 https://eur-lex.europa.eu/eli/reg/2016/1628/oj/eng).The generator, powered by a 12 HP, 4-stroke diesel engine, is designed to operate with standard diesel fuel. It was modified to allow connection to a second tank containing HVO. Between the use of different fuels, the system was properly heated to ensure there was no contamination or overlap of fuels in the production of soot particles. The injection of soot particles with the engine lasted only a few seconds to avoid exceeding the concentration of particles inside the chamber.


**Table 2 Combustion parameters selected for MISG propane combustion.**

| Labels of experiments | Air Flow [L min⁻¹] | Fuel Flow [mL min⁻¹] | Global equivalent ratio |
|---|---|---|---|
| P1 – P2 | 10 | 80 | 0.200 |
| P9 – P10 | 7 | 80 | 0.278 |

## 2.2 Experimental protocol

At the beginning of each experiment, soot particles were injected into ChAMBRe, and sampling began once the soot concentration stabilized. This typically occurs in approximately 3 minutes, corresponding to the chamber mixing time.





Once injected, the soot particles were left in suspension for defined timeframes and monitored with online instrumentation and sampled for offline analysis.

In this study, all the experiments were performed at atmospheric pressure, 21 °C< T <25 °C, RH<50 % and dark conditions.

## 2.3 Online optical aerosol measurements

One photoacoustic extinction meter (PAX) from Droplet Measurement Technologies was used to measure soot particles

absorption coefficients at 870 nm. PAX has two measurement cells where aerosol optical properties are determined by light absorption and scattering. Soot particles absorb light and release acoustic waves detected by a microphone and the intensity of the acoustic signal is interpreted to infer the particle absorption coefficient, while a wide-angle reciprocal nephelometer measures the scattering coefficient. No correction for truncation angle is applied, which can underestimate the scattering coefficient. However, since soot particles are generally smaller than 1 μm with SSA values below 0.3 (Moallemi et al., 2019),

this issue was disregarded.

One Giano_BC1 from Dadolab srl, a PMx sequential sampler with a built-in integrated Black Carbon optical monitor (Caponi et al. 2022) was connected to ChAMBRe, allowing for the continuous monitoring of BC concentrations on filter during the PM sampling. After sampling and the PM gravimetric determination, the same filter can be used for the thermal-optical OC and EC quantification. Thus, the MAC value (see paragraph 3.4 for more details) used to calculate BC concentrations can be

tuned to the specific composition of collected PM.

Data acquired during the experiments from different instruments were treated to homogenize their temporal resolution, in particular PAX data were averaged over the same 15-minute intervals used by Giano_BC1.

## 2.4 Offline aerosol measurements

Soot particles were collected for offline analysis on pre-baked quartz fibre filters (25, 37, 47-mm diameter, Grade Quartz, Heat

Treated Binderless Microfiber Filter, LabExact) using various low-volume samplers. The first one was the Giano_BC1, which was directly connected to ChAMBRe, without its sampling head. To prevent rapid depletion of the chamber, considering the numerous samplers and monitors connected to ChAMBRe during the experiments, the sampler was operated at the fixed flow of 10 L min$^{-1}$.

Additionally, up to four Gilian GilAir Plus personal samplers were used in parallel with different size-selective inlets: IOM

classifier for inhalable dust (up to 100 micrometers), two cyclones for respirable dust (up to 4 micrometers), and one personal impactor PEM-2-2.5 from TSI for particles less than 2.5 micrometers. These size-selective samplers adhere to health-based conventions adopted by ISO and CEN to define particle size-selective occupational exposure limits (OELs) for aerosols. These OELs match the relevant sites of aerosol deposition in the respiratory tract and the associated health effects for exposure assessment.

The four classifiers were directly inserted into the atmospheric chamber through a door flange located on one of the larger





flanges of the chamber central ring (Massabò et al. 2018). The samplers were secured inside the chamber using special hooks to keep them in the correct vertical position throughout the experiment (as if they were worn by an operator). The pumps were positioned outside the atmospheric chamber and connected to the dimensional selectors with specific tubes that passed through the door flange via small through-tube flanges.

All sampling began simultaneously with flow rates ranging from 1.7 to 10 L min$^{-1}$ for 60 to 190 minutes (see Table 1). The operating conditions and flow rates used with the samplers are reported in Table 3.

Particle-loaded filters were firstly analyzed using the multi-wavelength absorbance analyzer (MWAA; Massabò et al., 2013, 2015), a laboratory instrument for the offline direct quantification of aerosol absorption coefficients at five different wavelengths (850, 635, 532, 405, and 375 nm). These features have been utilized in several field campaigns in urban and rural

sites (Scerri et al., 2018; Massabò et al., 2019, 2020; Moschos et al., 2021) and in remote sites (Massabò et al., 2016; Saturno et al., 2017; Baccolo et al., 2020).

After MWAA measurements, the EC and OC mass concentrations were determined by thermal-optical transmittance analysis (TOT) using a Sunset Laboratory Inc. Sunset EC/OC analyzer and the NIOSH 5040 protocol (NIOSH, 2016). The NIOSH 5040 protocol is primarily intended for assessing workplace exposure to particulate diesel exhaust, but thermal-optical analysis

is also routinely applied to environmental carbonaceous aerosols (EN 16909:2017, Brown et al., 2017).

**Table 3 List of instruments and selectors used to collect aerosol particles during the experiments. Particle fraction collected, flow rate and filter diameter are also given.**

| Sampler | Classifier | Particles fraction collected | Flow rate [L min$^{-1}$] | Filter diameter [mm] |
|---|---|---|---|---|
| Gilian GilAir Plus | Cyclone | Respirable fraction | 1.7 | 25 |
| Gilian GilAir Plus | Cyclone | Respirable fraction | 1.7 | 37 |
| Gilian GilAir Plus | IOM | Inhalable fraction | 2 | 25 |
| Gilian GilAir Plus | TSI PEM-2-2.5 | PM$_{2.5}$ | 2 | 37 |
| Giano BC1 | None | Total | 10 | 47 |

**2.5 Size distribution measurements**

Particle concentration and size distribution inside the chamber were measured at 1-minute intervals using a scanning mobility particle sizer (SMPS 3938, TSI Inc.), equipped with a differential mobility analyzer (DMA 3081A) and a condensation particle counter (CPC 3750), operating at sheath/sample flow rates of 1.6/0.17 L min$^{-1}$. Measurements were corrected for diffusion losses using the instrument software. The SMPS was configured to measure particles with mobility diameters ranging from 18 to 806 nm.



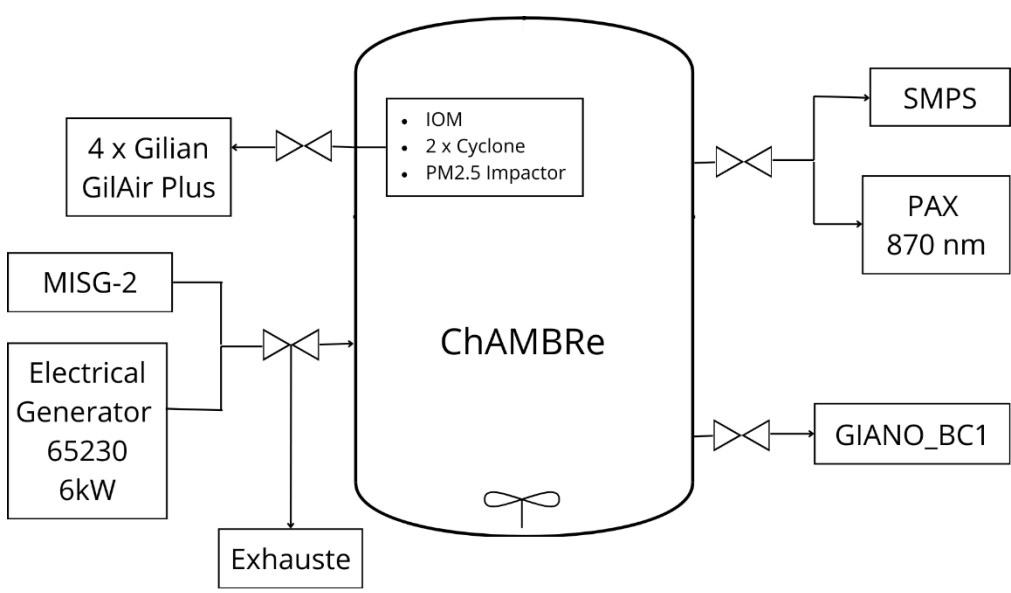


**Figure 1 Simplified layout of the experimental setup at ChAMBRe. The setup includes a mini-inverted soot generator (MISG-2), an electrical generator diesel engine (Electrical Generator 65230 – 6 kW), a Scanning Mobility Particle Sizer (SMPS), a Photoacoustic Extinctiometer (PAX) operating at 870 nm, a GIANO_BC1 sampler and up to four Gilian GilAir Plus personal samplers equipped with different size-selective inlets (IOM classifier for inhalable dust, cyclones for respirable dust and a PM$_{2.5}$ personal impactor for**
**airborne particles less than 2.5 μm).**

## 3. Results and discussion

### 3.1 EC/OC quantification

As shown in Figure 2, all the different types of size-selective samplers, which are designed to collect different size fractions of particulate matter to monitor worker exposure, showed compatible EC concentrations. This uniform efficiency across
different samplers and fuels, whether diesel, HVO, or propane, highlights their reliability in accurately measuring ultrafine particles, confirming that these classifiers are suitable for assessing worker exposure to soot particles, ensuring consistent and reliable data across various conditions and fuel types.

The EC:TC concentration ratio with propane resulted to be $(0.7 \pm 0.1)$, in accordance with the results published in Vernocchi et al., 2022.
The EC:TC concentration ratios were found to be EC:TC = $(0.15 \pm 0.05)$ and $(0.4 \pm 0.2)$, for diesel and HVO, respectively. OC was the dominant fraction in all samples except for those from the soot generator, where EC was dominant. Several studies have indicated that EC is the dominant component in PM emissions from diesel vehicles (Chiang et al., 2012; Grieshop et al., 2006; Kleeman et al., 2000), while other studies have reported contrasting results (Shah et al., 2004; Wu et al., 2016; Wang et al., 2021).





In this study, a high proportion of OC was observed, summing up 60% to 85% of TC in both HVO and diesel cases. The EC:OC ratio value is influenced by factors like emission standards, engine power, maintenance, fuel's chemical composition, physical properties, and experimental conditions (Lu et al., 2012; Zhang et al., 2009; Zhang et al., 2015). For example, Gali et al. 2017 indicated that under cold idle or low-engine-speed conditions, OC is the dominant fraction in particulate matter (PM), mainly originating from unburned fuel and incomplete combustion (Shah et al., 2004). Lower engine temperatures during

idling performed in this case study can result in less complete combustion, which likely explains the high OC levels observed. This high proportion of OC could be attributed to factors like incomplete combustion and lower engine temperatures during idling. Moreover, the heterogeneous effects of biodiesel on OC and EC may alter the composition of TC emissions (Williams et al., 2012; Agarwal et al., 2013). In addition, it should be noted that the determination of OC using a quartz filter can be affected by both positive and negative artifacts, such as volatilization losses and adsorption of vapor-phase organic compounds

(Eatough et al. 1993, Appel et al., 1983).

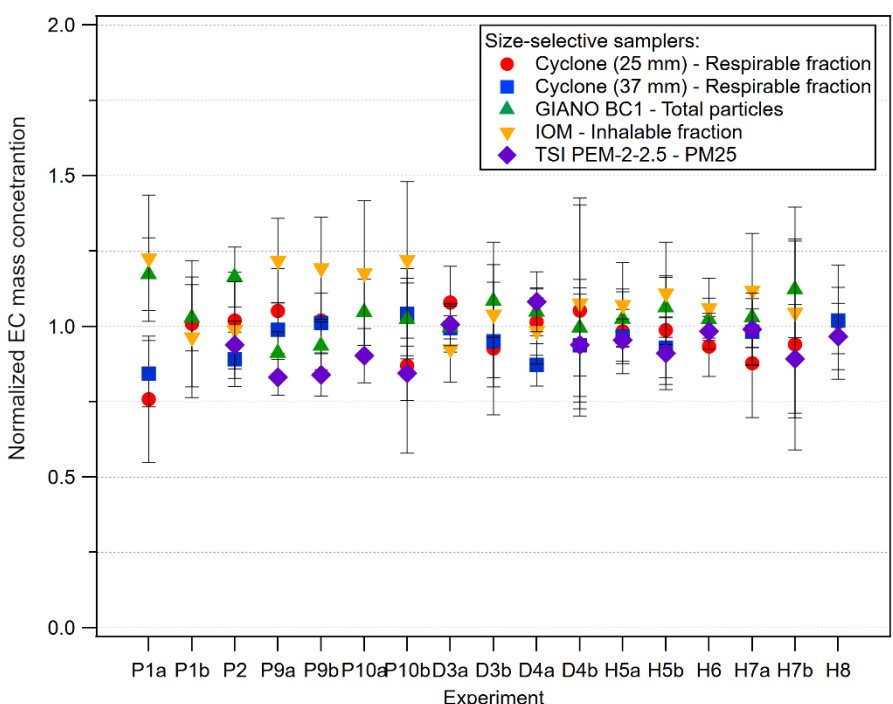

**Figure 2 Normalized EC mass concentration across different experiments. All selectors operated simultaneously within the same time period. In order to help in the interpretation of data values have been normalized to a scaling factor equal to the mean value of each dataset (i.e. mean values of EC mass concentration retrieved from filters analysis during each experiment).**

**3.2 Size distribution measurements**

A comparison of the size distributions of the generated aerosols for all fuels at the time of injection is presented in Figure 3. Data acquisition started 3 minutes after the injection of combustion aerosols and is reported here as the average over the

<br />
https://doi.org/10.5194/egusphere-2025-1447<br />



following 4 minutes time interval. For fresh soot immediately after injection into ChAMBRe, the size distributions for each fuel exhibit monomodal log-normal distributions. The MISG propane-soot aerosol shows a main peak between 200-300 nm, as expected (see Vernocchi et al., 2022). Regarding diesel exhaust emissions, particles from regular fossil diesel display a main peak roughly between 70 and 80 nm, while particles from HVO combustion show a slightly shifted peak at 80-90 nm.

The differences in the size distributions of aerosols generated from different fuels could be explained by the distinct combustion characteristics and chemical compositions of each fuel. For propane, the main peak between 200-300 nm is consistent with the combustion conditions (i.e. global equivalent ratio) used. The size distribution of the soot particles generated by the MSIG is mainly affected by the global equivalence ratio, with a general trend suggesting that by decreasing air flow rate the mode diameter of generated particles increased (Vernocchi et al., 2022 and reference therein).

In contrast, regular fossil diesel and HVO, which have a more complex hydrocarbon structure, tend to produce smaller particles. The main peak in the accumulation mode, consistent with literature data (Zhu et al., 2010, Chiavola O. et al., 2024, Böhmeke, C. et al., 2024), can be attributed to the incomplete combustion of heavier hydrocarbons, leading to the formation of smaller soot particles. Although here the two fuels show a similar size distributions, regular diesel generally emits more particles, with a size distribution shifted towards larger sizes compared to HVO. This is likely due to HVO's different chemical composition and aromatic-free nature, which may inhibit particle growth during combustion (Di Blasio et al., 2022). Overall, literature indicates that results are highly dependent on engine architecture and loads, injection type, operating conditions, and specific fuel properties like viscosity, density and oxygen content (Böhmeke et al., 2024, Chiavola et al., 2024 and reference therein). An accurate analysis of such aspects goes beyond the scopes of the present study.





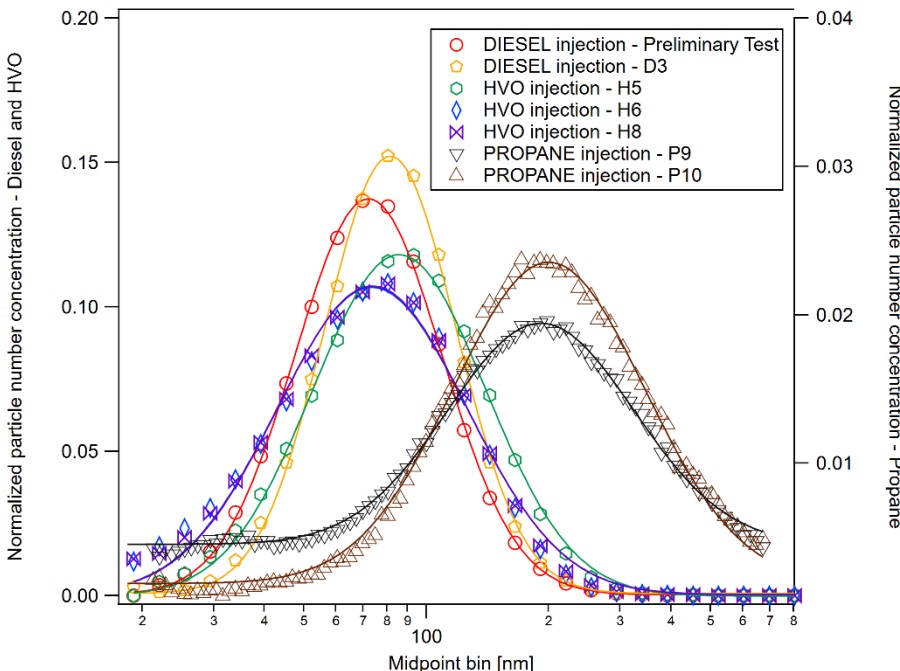

**Figure 3 Number size distribution measured (markers) and fit extrapolated log-normal size distribution (lines) of generated aerosols just after the injection in ChAMBRe. Size data are normalized by the total number for each distribution.**

## 3.3 Optical properties

The optical properties of the aerosol produced from each fuel were characterized by determining the absorption coefficient ($b_{abs}$). The $b_{abs}$ definition applies both to measurements directly performed on the aerosol dispersed in the atmosphere and to offline analysis on aerosol collected on filters, if appropriate data processing methods are applied (Massabò and Prati, 2021; and references therein). The $b_{abs}$ values were calculated offline by the MWAA analysis on the sampled filters during each experiment (see Sect. 2.4) and online by Giano_BC1 and the PAX. This gave the possibility to compare different optical

techniques on the same carbonaceous aerosol.

Offline MWAA analysis determined $b_{abs}$ at 635 and 850 nm on the sampled filters (see Sect. 2.4). The $b_{abs}$ values at 635 and 870 nm were derived also from online measurements throughout each experiment from Giano_BC1 measurements and PAX monitor, respectively.

The $b_{abs}$ values thus derived at different wavelengths, along with the elemental carbon (EC) concentration measured on the

filters (see sec. 2.4), were used to calculate the mass absorption coefficient (MAC) of the aerosol using the relation:

$$b_{abs} = MAC \times [EC] \, , \tag{3}$$

where $b_{abs}$ (Mm$^{-1}$) is the absorption coefficient, MAC (m$^2$ g$^{-1}$) is the mass absorption coefficient, and EC (µg m$^{-3}$) is the elemental carbon concentration.



The MAC values were calculated at 635 and 850/870 nm for each fuel, allowing the comparison at the same wavelength

(nearly) between the filter-based technique (i.e. the MWAA, see Figure 4) and the $b_{abs}$ retrieved online (at 635 nm from
Giano_BC1 and at 870 nm from PAX). All the measured MAC values are summarized in Table 4. The uncertainties were
estimated from the fit uncertainty (statistical error based on the uncertainty of optical coefficients and mass concentration). At
635 nm, data indicate that the two techniques are generally consistent, except for the results for propane, where the two methods
yielded significantly different MAC values. This discrepancy is likely due to the limited amount of data available for this fuel,

280    which is also reflected in the larger error margin associated with the MAC value. At 850/870 nm MWAA and PAX
measurements returned compatible MAC values for all the fuels. The results obtained with propane are in accordance with the
results published in Vernocchi et al., 2022.

**Table 4 Summary of the measured MAC values ($m^2$ $g^{-1}$).**

|  | Propane – MISG | DIESEL - 65230 – 6 kW -Hyundai | HVO - 65230 – 6 kW - Hyundai |
| --- | --- | --- | --- |
| **MAC values ($m^2$ $g^{-1}$) – MWAA 635 nm** | $6.1 \pm 0.2$ | $9.4 \pm 0.3$ | $8.0 \pm 0.3$ |
| **MAC values ($m^2$ $g^{-1}$) – Giano BC1 635 nm** | $7.8 \pm 1.1$ | $9.4 \pm 0.4$ | $8.4 \pm 0.6$ |
| **MAC values ($m^2$ $g^{-1}$) – MWAA 850 nm** | $5.2 \pm 0.5$ | $6.8 \pm 0.2$ | $6.0 \pm 0.3$ |
| **MAC values ($m^2$ $g^{-1}$) – PAX 870 nm** | $5.5 \pm 0.1$ | $6.2 \pm 0.5$ | $5.8 \pm 0.2$ |





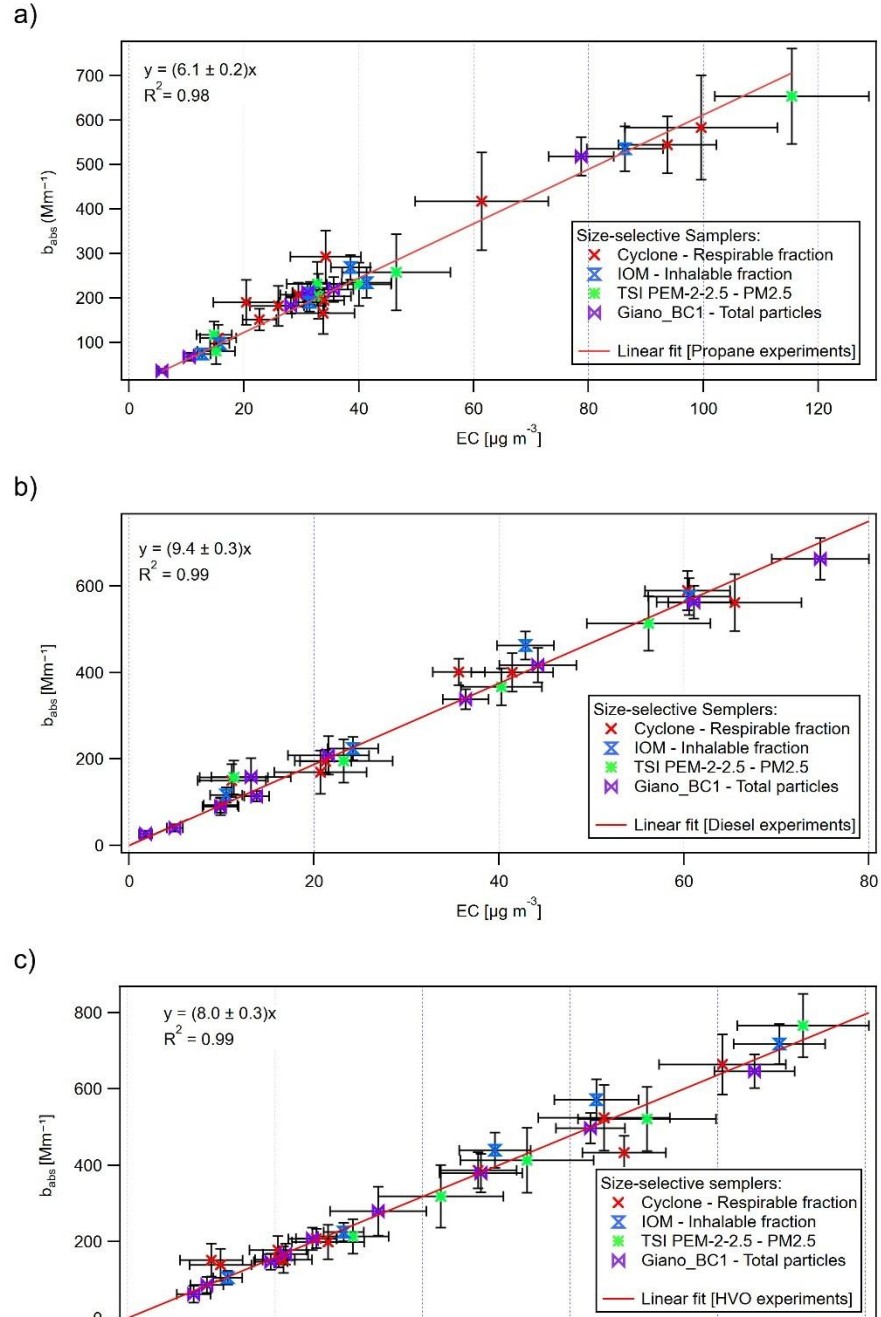

**Figure 4 Absorption coefficient (b$_{abs}$) at 635 nm, measured by MWAA vs. EC concentration (µg m$^{-3}$) for propane flame particles (a), commercial diesel exhaust particles (b) and HVO exhaust particles (c). For each classifier the corresponding particle size-selective sampling convention is also indicated. Linear fits (red lines) are plotted, with slopes representing the MAC values. The correlation coefficients (R²) for each fit are also provided.**





290 In general, the optical properties of the investigated aerosols, in terms of $b_{abs}$ and MAC, revealed differences in absorption characteristics across different fuels: particles generated by diesel combustion were found to be more light absorbing than those produced by propane and HVO. The MAC parameter values were higher for diesel, indicating more absorbent particulate matter.

Previous studies in the literature have shown that optical properties, such as absorption, depend on various factors including 295 composition, mixing state, aging, and size (Kirchstetter et al., 2004; Lewis et al., 2008; Lack et al., 2012; Lack and Langridge, 2013; Filep et al., 2013; Utry et al., 2013, Utry et al., 2014). Although the present study analyzed fresh soot particles, differences in composition can be expected, as the EC particles investigated in this study were generated by burning different fuels and through different combustion processes. This also resulted in differences in the size distribution of the particles produced (Figure 3), all of which can contribute to the variations observed in the MAC values.

300 **4. Conclusions**

The emissions of three different fuels combustion - propane, conventional fossil diesel, and Hydrotreated Vegetable Oil (HVO)- in terms of particle size distribution, optical properties, and EC concentration in the engine exhaust emissions were investigated using an atmospheric simulation chamber (ChAMBRe).

Soot particles were generated using a mini-inverted soot generator fuelled with propane and a diesel engine running on regular 305 diesel and Hydrotreated Vegetable Oil (HVO).

The study successfully quantified the EC/OC composition. Different types of size-selective samplers, designed to collect various particulate matter sizes for monitoring worker exposure, were tested, showing consistent EC concentrations across different fuels combustion (diesel, HVO, propane). This uniform efficiency confirms their suitability for assessing worker exposure to soot particles, ensuring consistent and reliable data across various conditions and fuel types. The EC:TC 310 concentration ratios were consistent with previous studies and indicated a higher proportion of organic carbon (OC) in diesel and HVO samples, highlighting the influence of several factors such as combustion condition and fuel composition on these ratios.

Size distribution measurements provided insights into the particle size distributions for different fuels, showing monomodal log-normal distributions with peaks varying based on the fuel type and combustion process. As indicated in previous studies, 315 the size distribution depends on different factors such as engine type and load, operating conditions, and fuel properties. In this case, fresh soot particles from propane showed a peak size distribution between 200-300 nm while diesel and HVO tend to produce smaller particles with a main peak in the accumulation mode.

Finally, the optical properties of the aerosols, in terms of absorption coefficient ($b_{abs}$) and mass absorption coefficient (MAC), varied significantly depending on the type of fuel. Particles generated by regular fossil diesel combustion were found to be 320 more light absorbing than those produced by propane and HVO, exhibiting higher MAC values. The MAC values, measured at different wavelengths (850/870 and 635 nm), ranged from 6.2 ± 0.5 to 9.4 ± 0.4 $m^2$ $g^{-1}$ for commercial diesel, from 5.2 ±

0.5 to 7.8 ± 1.1 m$^2$ g$^{-1}$for propane, and from 5.8 ± 0.2 to 8.4 ± 0.6 m$^2$ g$^{-1}$for HVO. Additionally, it should be noted that different optical analyses performed demonstrated compatible results in nearly all cases.

Overall, the findings underscore the importance of considering various factors in the assessment of carbonaceous aerosols
emissions and their optical properties. In particular, the variations observed on the mass absorption coefficient of particles produced under different combustion conditions highlight the importance of a deep understanding of such aspects.

Furthermore, instruments based on the measurements of optical properties due to their ability to provide continuous and real-time data, represent one of the most promising techniques for monitoring carbonaceous aerosols in both ambient air and workplace environments. Our findings indicate that aerosols produced by various combustion processes can have significantly
distinct optical properties, thus requiring an accurate evaluation of the correction factors adopted in optical measurements to obtain consistent and valid data under different conditions.

**Data availability**

The dataset for this paper can be accessed at https://data.mendeley.com/datasets/v6p5r5dmdy/1 (Danelli., 2025)

**Author contribution**

PB conceived the study. PB, SD, and LC designed the experiments and discussed the results. SD and LC conducted the experiments with contributions by VV, FM, MB and DM. LC performed the MWAA measurements. LC, MDC and MS performed the thermal–optical measurements. SD and LC performed the full data analysis under the supervision of PB and DM and with contributions from VV, MB and TI. FT and AP contributed to definition of experimental setup and data analysis procedure definition. SD, PB and PP wrote the manuscript. All authors reviewed and commented on the paper.

**Competing interests**

The authors declare that they have no conflict of interest.

**Acknowledgements**

The PM_TEN group would like to express its gratitude to the INFN technical staff affiliated with ChAMBRe for their valuable support.



**Funding**

This research has been supported by the CALIPSO project (Airborne Carbon: Limits, Impact, Protocols, and Operational Standards), funded by the Regional Program PR FESR 2021 – 2027 of the Liguria Region and the IR0000032–ITINERIS, Italian Integrated Environmental Research Infrastructures System (D.D. n. 130/2022 - CUP B53C22002150006) funded by the EU (Next Generation EU PNRR, Mission 4 "Education and Research", Component 2 "From research to business", Investment 3.1, "Fund for the realization of an integrated system of research and innovation infrastructures"). This study was carried out within the RETURN Extended Partnership and received funding from the European Union Next-GenerationEU (National Recovery and Resilience Plan – NRRP, Mission 4, Component 2, Investment 1.3 – D.D. 1243 2/8/2022, PE0000005)

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
