# Peer review of "Measurement report: Investigation of Optical Properties of Carbonaceous Aerosols from the Combustion of Different Fuels by an Atmospheric Simulation Chamber"

_EGUsphere, 2025_

## Referee Comment (RC2)

*Measurement report: Investigation of Optical Properties of Different Fuels Diesel Exhaust by an Atmospheric Simulation Chamber experiment* by Danelli et al., 2025 report the results of a study conducted to investigate the optical properties of aerosol produced from various diesel fuels. The experiments conducted were well designed and the resulting findings fit the scope of ACP. Results are of interest and demonstrate the effectiveness of using optical properties in evaluating EC; however, there are multiple instances where I could not follow the author's line of thought. The introduction and methods are well-written, with appropriate references to related literature. The results/discussion/conclusion as currently written do not adequately describe the measurements performed or aspects of how these measurements compare with existing literature. I favor publication after major revisions. I encourage the authors to carefully address the comments listed below to produce a revised manuscript for resubmission.
* * *
Major Comments

C1.  L306 – *"The study successfully quantified the EC/OC composition."* Results and explicit discussion of EC:OC is lacking. A figure/table summarizing the analysis of the EC:OC (or EC:TC) ratio for the carbonaceous aerosol produced would strengthen the measurements rather that just a description of the mass concentrations of TC. Are their more results from the Sunset EC/OC analyzer to assist with this that are not currently included?

C2.  Section 3.1 – Accordingly, with C1, the inclusion of a more thorough description/quantification for EC:OC is warranted.

C3.  L77 – It was mentioned that a goal of this study/grant was to evaluate and compare different methods of sampling/analyzing carbonaceous aerosols. The paragraph beginning with 208 explains this well and Figure 2 effectively shows the size-selective samplers performed consistently. I'd encourage a similar description in the conclusions to emphasize this finding
* * *
Minor/Technical Comments

C4.  Throughout the manuscript there are odd paragraph breaks that produce a confusing structure when reading. Specific examples are listed below.

C5.  Currently, the title does not accurately summarize the contents. Maybe: *"Measurement Report: Investigation of the Optical Properties of Carbonaceous Aerosols Produced by Diesel Fuel Combustion in an Atmospheric Simulation Chamber"*.

C6.  L29 – The use of Indeed would better indicate agreement with the previous statement instead of Actually

C7.  L30 – Replace clime with climate

C8. L35 – The use of Additionally would be a better adverb to include additional information to the previous statement.

C9. L79 – The use of alternatively here is confusing.

C10. L83 – Replace physic-chemical with physicochemical

C11. L88-L94 – I would move the citations of the full description of the chamber to the end of the brief description included.

C12. L96-L97 – I would try listing *"…online and offline gaseous composition and aerosol concentration and properties inside the volume:"* differently so the use of and isn't repeated. Maybe *"…online and offline gaseous composition in addition to aerosol concentration and properties inside the volume:"*

C13. L110-L112 – Confusing use of alternatively. Please reword this sentence to describe the two combustion techniques and their associated fuel type.

C14. L127-L128 – Paragraph break here is odd

C15. L128 – Move this sentence to the end of the paragraph ending on L120 so the description of MISG is condensed.

C16. Sect. 2.2 – All sentences within this section should be one paragraph.

C17. L159 – Rather than within the results, the method to calculate MAC value would be better placed in the methods.

C18. Table 3 – Change Particles in the third column header to Particle

C19. L209 – "… showed comparable EC concentrations."

C20. Figure 2 – Some x-axis labels overlap.

C21. L237 – The description of data acquisition for the SMPS should be in methods not results.

*C22.* L246 – "*…increased (Vernocchi et al., 2022 and references therein).*"

*C23.* L254 – Similar to C21 references should replace reference

*C24.* Figure 3 – Specify the fits are for a monomodal log-normal size distribution.

*C25.* L324 – Explicitly list the factors of importance

*C26.* L237 – This sentence could be improved with some restructuring. Needed to read it a couple times to understand it.

---

## Author Comment (AC1)

**Anonymous Referee #1, 17 Apr 2025**

The study presented here is a valuable contribution to the aerosol community. It provides optical properties, more specifically mass absorption cross section values, for three different soot-like aerosol samples. Two of the samples are based on the combustion of liquid fuels burned in a real diesel engine. The manuscript is well written and the data are well presented. I would suggest publishing it after addressing the following comments:

**Scientific comments**

- Provide information on the calibration of the various instruments, especially the PAX.

The SMPS and Giano BC1 instruments undergo factory calibration on an annual basis. The PAX was calibrated following the procedures outlined in the user manual, immediately prior to the start of the experimental campaign. The personal samplers were calibrated using a certified flowmeter before each experiment to verify and adjust the sampling flow, in accordance with the NIOSH Manual of Analytical Methods (NMAM), 5th Edition.

We believe that, while important, these technical details are not essential to include in the main manuscript.

- The EC:TC ratio is quite different for the different samples. The larger amount of OC in some samples could affect the MAC amid coating. However, the authors do not address this issue. Please comment on this.

We thank the Reviewer for this observation. We have addressed this point in the revised manuscript at line 335, where we added the following sentences:

*"MAC parameter values are higher for diesel, indicating more absorbent particulate matter, while HVO and propane show lower MAC values, even above 20%. This behavior is consistent with the EC:TC ratios shown in Table 4: the presence of OC coating soot particles enhances light absorption through the lensing effect (Bond et al., 2006; Lack et al. 2010; Lefevre et al., 2018)."*

We added the relative references in the corresponding section.

*Bond, T. C., Habib, G., and Bergstrom, R. W.: Limitations in the enhancement of visible light absorption due to mixing state, J. Geophys. Res. Atmospheres, 111, 2006JD007315, https://doi.org/10.1029/2006JD007315, 2006.*

*Lack, D. A. and Cappa, C. D.: Impact of brown and clear carbon on light absorption enhancement, single scatter albedo and absorption wavelength dependence of black carbon, Atmospheric Chem. Phys., 10, 4207–4220, https://doi.org/10.5194/acp10-4207-2010, 2010.*

*Lefevre, G., Yon, J., Liu, F., and Coppalle, A.: Spectrally resolved light extinction enhancement of coated soot particles, 955 Atmos. Environ., 186, 89–101, https://doi.org/10.1016/j.atmosenv.2018.05.029, 2018*

- It has been shown that MISG soot has an average diameter between 200-300 nm. The very different particle diameter compared to the other two samples will strongly affect the MAC values, making these results not comparable. In addition, the large size of MISG-generated soot particles is not representative of engine exhaust soot. Is the use of the MISG still justified?

The choice to include the MISG in this study was motivated by its ability to generate almost pure elemental carbon (EC) particles. Inverted-flame burners are widely regarded as ideal soot sources due to their capability to operate under controlled fuel-lean conditions and to produce mature, EC-dominated soot. We adopted this configuration using MISG as a reference source.

To clarify this point, we have added the following sentences to the manuscript:

- Line 121: "*The MISG is an inverted-flame burner often considered an ideal soot source due to its capacity to generate almost pure EC particles (Stipe et al., 2005; Moallemi et al., 2019, and references therein).*"

- Line 126: "*In this study, the MISG, considered a reference EC-dominated soot source, was fueled with propane at a fixed air-to-fuel ratio, following Vernocchi et al. (2022).*"

- It would be interesting to see how the different absorption coefficient measurement techniques compare in terms of b_abs.

We thank the Reviewer for the suggestion. To address this point, we report here the correlation analysis between the b_abs values obtained from the different instruments used in the study.

[Figure]

The results show good agreement between techniques, with correlation coefficient ($R^2$) above 0.9 and slopes close to unity (within uncertainty). These findings are also in line with previous works, such as Vernocchi et al., 2022 and Caponi et al., 2022, were similar comparisons yielded comparable results, as also referenced in the manuscript.

To keep the main text focused, we decided to not include the correlation plots in the manuscript, but we would be happy to provide them as supplementary material if the reviewer or editor considers it useful.

- In Table 4, it would be valuable to include the MAC values interpolated to 550 nm in addition to those already shown. This would make it easier for the reader to make comparisons with literature values.

We thank the Reviewer for the valuable suggestion. Considering also the related comment below regarding the absorption Ångström exponents, we have added the MAC values extrapolated to 550 nm in Table 4 (now Table 5). These values were calculated using the AAE derived from MWAA measurements, as described in the newly added text at line 308.

- What is the wavelength in Figure 4? Please add it to the axis labels.

The wavelength is 635 nm as reported in the figure caption. We have now added this information in the y-axis label for clarity.

- I would recommend showing the absorption angstrom exponents measured by the MWAA.

As mentioned in our response above, we have now included the absorption Ångström exponents measured by the MWAA. The following sentences have been added at line 308:

*"An average Ångström absorption exponent (AAE; Moosmüller et al., 2011) was calculated for each fuel by aggregating all available b_abs datasets from the MWAA analysis (Table 5). To facilitate comparison with literature data, the MAC value at 550 nm was extrapolated from the MWAA measurement using the relation $MAC_{550} = MAC_{635} (635/550)^{AAE}$. All the measured MAC values are summarized in Table 6."*

- The conclusion section is weak. There is no discussion of the implications of the MAC values found in the study. Please improve.

We thank the Reviewer for this important observation. Since all reviewers suggested to improve the conclusion section at different point, we have revised and expanded it to better summarize the key findings of the study and to discuss their broader implications. The Conclusion Section is now rewritten as follows:

[revised manuscript text omitted]

- "Correction factors" are mentioned on page 16. Please elaborate. What corrections are referenced?

Thank you for the comment. We agree that the term "correction factors" was unclear and potentially misleading. We have reformulated the concept as follow:

*"However, the significant differences in aerosol optical properties across combustion processes require an accurate source characterization in order to apply the most appropriate MAC values when interpreting data from optical instruments."*

**Typos**

- Page 11, second paragraph: MISG is misspelled.

Word corrected.

---

## Author Comment (AC2)

**Anonymous Referee #2, 28 Apr 2025**

Measurement report: Investigation of Optical Properties of Different Fuels Diesel Exhaust by an Atmospheric Simulation Chamber experiment by Danelli et al., 2025 report the results of a study conducted to investigate the optical properties of aerosol produced from various diesel fuels. The experiments conducted were well designed and the resulting findings fit the scope of ACP. Results are of interest and demonstrate the effectiveness of using optical properties in evaluating EC; however, there are multiple instances where I could not follow the author's line of thought. The introduction and methods are well-written, with appropriate references to related literature. The results/discussion/conclusion as currently written do not adequately describe the measurements performed or aspects of how these measurements compare with existing literature. I favor publication after major revisions. I encourage the authors to carefully address the comments listed below to produce a revised manuscript for resubmission.

**Major Comments**

C1. L306 – "The study successfully quantified the EC/OC composition." Results and explicit discussion of EC:OC is lacking. A figure/table summarizing the analysis of the EC:OC (or EC:TC) ratio for the carbonaceous aerosol produced would strengthen the measurements rather that just a description of the mass concentrations of TC. Are their more results from the Sunset EC/OC analyzer to assist with this that are not currently included?

We thank the Reviewer for the comment. We agree that a more detailed presentation and argumentation of the EC:OC composition would enhance the robustness of the manuscript. We have now added a new table in Section 3.1 (Table 4), which summarizes the EC:OC and EC:TC ratios for each fuel type analyzed. Furthermore, we have expanded the discussion in the Conclusion section to emphasize the variability in carbonaceous aerosol composition across different combustion sources and its implications. The modified Conclusion is reported in C3 answer. We hope this addition provides a clearer picture of the carbonaceous aerosol composition and better supports the interpretation of the optical properties presented in the study.

C2. Section 3.1 – Accordingly, with C1, the inclusion of a more thorough description/quantification for EC:OC is warranted.

As mentioned in our response to Comment C1, we have revised Section 3.1 to include a table summarizing the EC:OC and EC:TC ratios for each fuel. In addition, we have reviewed the Section 3 and especially the Conclusion section to make the discussion on the EC:OC ratio more explicit and to better highlight the implications of these findings.

C3. L77 – It was mentioned that a goal of this study/grant was to evaluate and compare different methods of sampling/analyzing carbonaceous aerosols. The paragraph beginning with 208 explains this well and Figure 2 effectively shows the size-selective samplers performed consistently. I'd encourage a similar description in the conclusions to emphasize this finding

We appreciate the Reviewer's suggestion. In the revised Conclusion section, we have included a dedicated paragraph to underscore this important aspect of the study. Now the conclusion was modified as follows:

[revised manuscript text omitted]

**Minor/Technical Comments**

C4. Throughout the manuscript there are odd paragraph breaks that produce a confusing structure when reading. Specific examples are listed below.

C5. Currently, the title does not accurately summarize the contents. Maybe: "Measurement Report: Investigation of the Optical Properties of Carbonaceous Aerosols Produced by Diesel Fuel Combustion in an Atmospheric Simulation Chamber".

We thank the Reviewer for the suggestion. We have modified the Title in:

*"Measurement Report: Investigation of Optical Properties of Carbonaceous Aerosols from the Combustion of Different Fuels by an Atmospheric Simulation Chamber."*

C6. L29 – The use of Indeed would better indicate agreement with the previous statement instead of Actually

Thanks, done.

C7. L30 – Replace clime with climate

Thanks, done.

C8. L35 – The use of Additionally would be a better adverb to include additional information to the previous statement.

Thanks, done.

C9. L79 – The use of alternatively here is confusing.

We have reformulated the sentence as: "*Independent experiments were conducted inside an atmospheric simulation chamber (ASC) connected first to a soot generator and then a commercial diesel engine running on regular diesel and Hydrotreated Vegetable Oil (HVO).*"

C10. L83 – Replace physic-chemical with physicochemical

Thanks, done.

C11. L88-L94 – I would move the citations of the full description of the chamber to the end of the brief description included.

Thanks, done.

C12. L96-L97 – I would try listing "...online and offline gaseous composition and aerosol concentration and properties inside the volume:" differently so the use of and isn't repeated. Maybe "...online and offline gaseous composition in addition to aerosol concentration and properties inside the volume:"

Thanks, done.

C13. L110-L112 – Confusing use of alternatively. Please reword this sentence to describe the two combustion techniques and their associated fuel type.

Thanks, done

C14. L127-L128 – Paragraph break here is odd

Thanks, done

C15. L128 – Move this sentence to the end of the paragraph ending on L120 so the description of MISG is condensed.

Thanks, done

C16. Sect. 2.2 – All sentences within this section should be one paragraph.

Thanks, done

C17. L159 – Rather than within the results, the method to calculate MAC value would be better placed in the methods.

We thank the Reviewer for this suggestion. We have created a new subsection within the Materials and Methods section, titled "Retrieval of aerosol mass absorption cross section" (Section 2.5), where we have relocated the description of the MAC calculation.

C18. Table 3 – Change Particles in the third column header to Particle

Thanks, done

C19. L209 – "... showed comparable EC concentrations."

Thanks, done.

C20. Figure 2 – Some x-axis labels overlap.

Thank you, Figure 2 has been revised accordingly.

C21. L237 – The description of data acquisition for the SMPS should be in methods not results.

We thank the Reviewer for the observation. In this case, the sentence in question introduces Figure 3 and contextualizes the data shown. For this reason, we preferred to leave it here.

C22. L246 – "...increased (Vernocchi et al., 2022 and references therein)."

Thanks, done

C23. L254 – Similar to C22 references should replace reference

Thanks, done.

C24. Figure 3 – Specify the fits are for a monomodal log-normal size distribution.

Thank you, done.

C25. L324 – Explicitly list the factors of importance

Thanks for the comment; we have reformulated the paragraph as follows:

*In conclusion, the findings underscore the importance of considering several factors in the assessment of carbonaceous aerosols emissions and their optical behavior. The type of emission source (e.g. engine type), the chemical composition of the fuel, and the specific combustion condition (e.g. temperature, efficiency) influence the optical properties of the emitted particles. In particular, the variability of the mass absorption coefficient under different combustion scenarios highlights the importance of a deep characterization of such aspects.*

C26. L237 – This sentence could be improved with some restructuring. Needed to read it a couple times to understand it.

We reformulated the sentence as follows:

"*The data acquisition started 3 minutes after the end of injection of combustion aerosols; the data reported here are the average of the 4 consecutive minutes time interval*".

---

## Author Comment (AC3)

**Anonymous Referee #3, 14 May 2025**

Danelli et al. present a study investigating the EC/OC ratio, particle size distribution, and MAC of carbonaceous aerosols produced from the combustion of various fuels using an atmospheric simulation chamber. The experimental setup is well designed, and the results provide meaningful insights into the optical properties of combustion-derived aerosols. This study aligns with the scope of ACP as a measurement report. I recommend it for publication, provided the following comments are addressed:

1. The current title may be misleading, as not all particles investigated are derived from diesel exhaust. I suggest revising the title to better reflect the scope of the study. For example:

   "Investigation of Optical Properties of Carbonaceous Aerosols from the Combustion of Different Fuels Using an Atmospheric Simulation Chamber."

   We thank the Reviewer for the suggestion. We have modified the Title in:
   *"Measurement Report: Investigation of Optical Properties of Carbonaceous Aerosols from the Combustion of Different Fuels by an Atmospheric Simulation Chamber."*

2. While the study focuses on aerosol optical properties, no health-related outcomes are presented. Therefore, I suggest removing or minimizing health-related discussions in the Introduction, as they may distract from the main focus.

   Thanks for the comment, we have reviewed the introduction as suggested.

3. For the MISG experiments, please clarify the rationale for selecting the specific "global equivalence ratio." Was this condition intended to replicate soot production mechanisms similar to those in diesel engines? Since the EC/OC ratio is highly sensitive to combustion stoichiometry, a justification for this choice is essential to contextualize comparisons with engine-derived emissions. Additionally, please ensure the reference "Vernocchi et al., 2022" is listed in the References, as it appears to be missing.

   We thank the Reviewer for the comment. The MISG was included in this study due to its ability to generate nearly pure EC particles. Inverted-flame burners such as the MISG are widely recognized as ideal soot sources, as they operate under controlled conditions and produce mature, EC-dominated soot. In our study, we do not consider the MISG settings to represent any specific type of atmospheric soot; rather, MISG was used as a reference source under operating conditions selected following Vernocchi et al. 2022 (now added to the References section).

   To clarify this point, we have added the following sentences to the manuscript:

   - Line 121: "*The MISG is an inverted-flame burner often considered an ideal soot source due to its capacity to generate almost pure EC particles (Stipe et al., 2005; Moallemi et al., 2019, and references therein).*"

   - Line 126: "*In this study, the MISG, considered a reference EC-dominated soot source, was fueled with propane at a fixed air-to-fuel ratio, following Vernocchi et al. (2022).*"

4. The measured particle size distributions are based on mobility diameters (SMPS). Did the authors consider the influence of particle morphology? For instance, the larger mobility diameters observed for soot generated by MISG are likely due to the aggregate structure typical of high-EC (soot-rich) particles. This morphology can significantly inflate the mobility diameter relative to the volume-equivalent diameter and should be discussed when interpreting the size data.

We thank the Reviewer for this observation. We agree that particle morphology can influence size measurements. To address this point, we have added the following sentence to the manuscript (line 280):

*"In combustion processes where EC predominates, such as in the case of propane used in this study, the emitted particulate matter typically exhibits the characteristic fractal-like structure of soot, which may affect the electric mobility diameter measured by the SMPS, resulting in greater values of mobility diameters. In contrast, when the soot OC fraction increases, the particle size tends to decrease, and the morphology shifts toward more compact, rounded aggregates (Heuser et al., 2024, Leskinen et al., 2023)."*

*Heuser, J., Di Biagio, C., Yon, J., Cazaunau, M., Bergé, A., Pangui, E., Zanatta, M., Renzi, L., Marinoni, A., Inomata, S., Yu, C., Bernardoni, V., Chevaillier, S., Ferry, D., Laj, P., Maillé, M., Massabò, D., Mazzei, F., Noyalet, G., Tanimoto, H., Temime-Roussel, B., Vecchi, R., Vernocchi, V., Formenti, P., Picquet-Varrault, B., and Doussin, J.-F.: Spectral optical properties of soot: laboratory investigation of propane flame particles and their link to composition, EGUsphere [preprint], doi: 10.5194/egusphere-2024-2381, 2024.*

*Leskinen, J., Hartikainen, A., Väätäinen, S., Ihalainen, M., Virkkula, A., Mesceriakovas, A., Tiitta, P., Miettinen, M., Lamberg, H., Czech, H., Yli-Pirilä, P.: Photochemical Aging Induces Changes in the Effective Densities, Morphologies, and Optical Properties of Combustion Aerosol Particles. Environ. Sci. Technol. 57, 13, 5137–5148, 2023*

5. Please include the standard deviation of the EC:TC ratio in the abstract.

    Thanks for the comment, we have added the standard deviation in the abstract.